# Amuro & Char:
# Analyzing the Relationship between Pre-Training and Fine-Tuning of Large Language Models

## Abstract

Large language model development relies on the pre-train-then-align paradigm, in which the model is typically pre-trained on a large text corpus and undergoes a tuning stage to align the model with human preference or downstream tasks. We investigate the relationship between pre-training and fine-tuning by fine-tuning multiple intermediate pre-trained model checkpoints to understand how models develop as they train. Our results on 18 datasets suggest that i) continual pre-training improves the model in a latent way that manifests after fine-tuning; ii) fine-tuning most benefits datasets where the model does not show capability during pre-training; iii) although the model benefits significantly through supervised fine-tuning, it may forget previously known domain knowledge and tasks not seen during fine-tuning; iv) the model exhibits high sensitivity to evaluation prompts after supervised fine-tuning, but this sensitivity can be alleviated through more pre-training. [1]

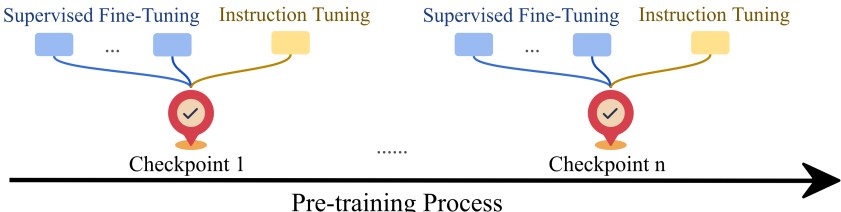

Figure 1: Illustration of the experimental scheme.

# 1 Introduction

The rise of large language models (LLMs) as a general-purpose tool for a diverse range of natural language processing tasks has dramatically transformed the field, introducing new paradigms for data collection and model training (Brown et al., 2020, Biderman et al., 2023, Touvron et al., 2023, Jiang et al., 2023, Chowdhery et al., 2023, Groeneveld et al., 2024, Wang et al., 2024, *inter alia*). Numerous models, training methods, datasets, and evaluation methods continue to be developed on an ongoing basis. Nevertheless, a unified paradigm has emerged for training LLMs: pre-train on an enormous corpus of diverse documents, ranging from 250B (Biderman et al., 2023) to 15T (AI@Meta, 2024) tokens, followed by an alignment stage to make the model more useful and performative for various tasks.

Based on this paradigm, work has focused on improving these two stages. Work to improve pre-trained models includes larger training sets (Hoffmann et al., 2022; AI@Meta, 2024; Touvron et al., 2023), different data selection mechanisms (Xia et al., 2024), higher quality data (Zhou et al., 2024),

---

[1]Code, results, and data to reproduce the experiments are available at https://anonymous.4open.science /r/AmuroCharRelease-DEC5. All the model checkpoints resulting from this work are available at [redacted for anonymity]

and various model architectures (Su et al., 2024; Touvron et al., 2023). Meanwhile, research on model alignment includes different training objectives (Rafailov et al., 2024; Schulman et al., 2017), new datasets (Narayanan & Aepli, 2024), more efficient training (Hu et al., 2021; Dettmers et al., 2024) and safety tuning (Bianchi et al., 2023). The alignment stage usually involves either supervised fine-tuning for specific tasks or instruction fine-tuning for general-purpose usage. Regardless, fine-tuning (almost always) comes at the end of pre-training and yields remarkable improvements on downstream tasks (Touvron et al., 2023; Groeneveld et al., 2024). Consequently, the benefits of each stage are largely explored independently, with improvements to pretraining being orthogonal to benefits from model alignment.

Rather than exploring these two training regimes independently, we ask: **How do pretraining and fine-tuning interact to produce the resulting model?** Does more pre-training hinder better fine-tuning results? What does the model learn and forget during pre-training and fine-tuning? Answering these questions requires us to examine how models learn during pre-training and how this affects fine-tuning. Therefore, we fine-tune **multiple pre-training checkpoints** of a large language model (Figure 1), evaluating each checkpoint and its fine-tuned variant on downstream evaluation sets. We track model abilities during pre-training and compare them to improvements achieved after fine-tuning at the corresponding pre-training step. We explore both supervised and instruction fine-tuning, testing the models' memorization and forgetting when learning specific tasks and serving as general-purpose language-AI tools. We believe that we are the first to explore the relationship between fine-tuning and pre-training by fine-tuning intermediate model checkpoints.

Our experiments yield the following insights into LLM training: we find that (1) continued pre-training can improve a model in ways that are only revealed after fine-tuning (§5); (2) tasks for which the model already performs well during pre-training benefit much less from fine-tuning than those where the model does not demonstrate capabilities (§4, §5); (3) although supervised fine-tuning can improve performance on in-distribution tasks, it can also cause the model to forget domain knowledge or tasks that it was previously capable of solving (§6); (4) fine-tuned models show high sensitivity to evaluation prompts, but this sensitivity can be alleviated by more pre-training (§6).

Our findings provide insights into model training and can inform methods for both pre-training and fine-tuning. Furthermore, our work shows the value of analyzing the training dynamics, in addition to analyzing the final checkpoint of an LLM, as an aspect of interpretability, and we encourage model developers to release these checkpoints to aid future studies.

## 2 BACKGROUND: MODEL TRAINING

We begin with a brief survey of the core components of LLM training: pre-training, fine-tuning, and instruction fine-tuning. We also discuss the related topic of in-context learning as well as different efficient fine-tuning strategies. We use "model alignment" as a general term for techniques that align a model with a desired behavior, which can be accomplished by fine-tuning models after pretraining. The term is also associated with other definitions (Shen et al., 2024). We also note several related studies that explore training dynamics to understand model behavior (Tirumala et al., 2022; Chen et al., 2023; Tian et al., 2023). With this in mind, we conduct an empirical study on how the amount of pre-training affects the effectiveness of fine-tuning.

**Pre-training.** The first step of training a LLM is pre-training on a massive text corpus (Achiam et al., 2023; Touvron et al., 2023; Groeneveld et al., 2024). For decoder-only models in the GPT family, the subject of our paper, work since the introduction of GPT-2 (Radford et al., 2019) has focused on scaling up model training. Initial work increased model size to hundreds of billions of parameters (Brown et al., 2020; Rae et al., 2021; Chowdhery et al., 2023), along with explorations in model size, training corpus size, and training data characteristics (Hoffmann et al., 2022; Gururangan et al., 2020). Since the push towards large models, work has shifted to increasing the amount of pre-training data (Computer, 2023; Soldaini et al., 2024), with new models now reaching 15 trillion tokens (AI@Meta, 2024). Studies of model performance on various tasks at different model sizes introduced the idea of emergent model abilities (Wei et al., 2022), with new model abilities being revealed as model training grows.

We also recognize a particularly important trend for this paper: model openness. Early LLMs were proprietary models accessible only through an API. The first large open model, Bloom (Bloom Ström

et al., 2023), allowed widespread LLM evaluation. Subsequent open models, such as OPT (Zhang et al., 2022), LLaMA (Touvron et al., 2023; Keles & Bayraklı, 2024) and others (Biderman et al., 2023; Gururangan et al., 2023b; Almazrouei et al., 2023), have become the norm. In this paper, we study OLMo (Groeneveld et al., 2024), one of the only models to release individual pre-training checkpoints.

**Fine-Tuning.** Early work on instruction fine-tuning using reinforcement learning with human feedback (RLHF) (Ziegler et al., 2019; Stiennon et al., 2020; Ouyang et al., 2022) demonstrates the dramatic effect that model alignment could have on a pre-training model. When a specific task of interest has been identified, supervised fine-tuning can improve a pre-trained model. Task-agnostic tuning became popularized with the advent of T5 models (Raffel et al., 2020), where a pre-trained LLM is tuned using a general text-to-text solution. When multiple tasks are given to the model, the model is commonly given a task-specific prefix or an instruction along with the task input, leading to the development of various methods of prefix tuning (Li & Liang, 2021) and instruction tuning (Wei et al., 2021; Mishra et al., 2022; Victor et al., 2022).

**Instruction Fine-Tuning.** Instruction fine-tuning is preferred when more general model behaviors are desired. Popularized through reinforcement-learning with human feedback (RLHF) (Christiano et al., 2017; Ziegler et al., 2019; Stiennon et al., 2020; Ouyang et al., 2022) and reinforcement-learning with AI feedback (RLAIF) (Lee et al., 2023), these methods utilize a reward model to simulate human feedback. Others explore human preference tuning without a reward model (Rafailov et al., 2024; Song et al., 2024; Xu et al., 2024), or study the effects of these tuning methods (Shen et al., 2024; Perez et al., 2023). Sharma et al. (2024) show that supervised fine-tuning can lead to similar performance as RLAIF.

**In-Context Learning.** ICL, also called few-shot learning, is also used as an evaluation strategy where the model is given a prompt composed of examples of tasks expected to be solved. The underlying model is evaluated based on its response to the input. ICL can benefit from a larger context window that adds more examples, which can spur work on the development of model quantization techniques (Dettmers et al., 2022) and the alleviation of hardware constraints (Brown et al., 2020; Xie et al., 2021; Min et al., 2022). While not the subject of this paper since it does not make changes to model parameters, in-context learning utilizes a small amount of supervised data to improve model performance.

**Fine-Tuning Techniques.** While model pre-training can be done by a few groups with large resources interested in developing new models, fine-tuning depends on the task and is of broad interest. Therefore, many techniques facilitate time-, memory-, and data-efficient model training through parameter-efficient fine-tuning (PEFT) (Hu et al., 2021; Liu et al., 2021; 2023), quantization (Jacob et al., 2018; Dettmers et al., 2022; 2024), and specialized data filtering (Xia et al., 2024; Zhou et al., 2024; Attendu & Corbeil, 2023). This paper focuses specifically on full-parameter fine-tuning, while our findings suggest the potential for data-efficient and budget-friendly training by understanding the critical turning point of model training. Our findings are closely related to the recent study on *phase transition* of model training (Olsson et al., 2022; Wei et al., 2022; Chen et al., 2023).

## 3 EXPERIMENTAL SETUP

In this section, we describe the models and datasets used. The hyperparameter tuning procedure and setup for each fine-tuning setting can be found in Appendix A.

### 3.1 MODEL CHOICE

Our paper primarily considers OLMo-1B (Groeneveld et al., 2024), a high-performing open-source large language model. Llama3-8B (AI@Meta, 2024) is also used to verify part of our findings that do not require access to the pre-training history. Ideally, we would evaluate multiple models, but OLMo is the only model to release intermediate pre-training checkpoints, and thus the only model

---

[2]https://huggingface.co/datasets/pietrolesci/gpt3_nli

| Supervised Fine-Tuning | | | |
|---|---|---|---|
| **Task** | **Training** | **ID Test** | **OOD Test** |
| Summary Generation | XSum | XSum, XLSum | CNN |
| Question Generation | SocialIQa | SocialIQA | SciQ, TweetQA |
| Natural Language Inference | MNLI | MNLI1, MNLI2 | RTE, GPT3NLI[2] |
| Paraphrase Detection | Paws | Paws | QQP, STS-B |
| **Instruction Tuning** | | | |
| **Dataset** | **Description** | | |
| TÜLU-v2 | A mixture of instruction datasets. | | |
| ARC | Grade-school multiple-choice QA. | | |
| OpenbookQA | Open book exam QA. | | |
| Hellaswag | Commonsense inference. | | |
| BoolQ | Reading comprehension. | | |
| SciQ | Science exam multiple choice QA. | | |

Table 1: Dataset information. For Generation tasks, ROUGE-L is used as the evaluation metric, and accuracy is used for classification tasks.

that supports our checkpoint analysis[3][4]. While we recognize the limitation of only having access to one model's pre-training checkpoints, our results provide needed evidence concerning model training behaviors.

Despite being the only open model with pre-training checkpoints, it has several desirable properties. First, the model is fully open, including the training details, pre-training data, and fine-tuning data. Second, the smaller model size allows us to train efficiently on a single A100 GPU without parameter-efficient training or quantization that introduce additional confounding factors. We also note that OLMo-1B compares favorably to its larger version, and recent work has shown that small models can compete with larger ones (Riviere et al., 2024).

We select model pre-training checkpoints uniformly from the pre-training history along with the first and the final checkpoints.

### 3.2 TRAINING PROCEDURE

We fully fine-tune each of the selected model checkpoints using two different procedures to create fine-tuned models: supervised fine-tuning and instruction tuning. The supervised fine-tuning is conducted separately for each model checkpoint and dataset, while the instruction fine-tuning is done once using the instruction dataset. The instruction-tuned model is evaluated on a suite of LLM benchmarks.

**Supervised Fine-tuning.** We adapt the datasets from Yang et al. (2024) for supervised fine-tuning. For each in-domain dataset, one to two cross-domain evaluation datasets are supplied. Each pre-training checkpoint of OLMo is fully fine-tuned for 3 epochs with a batch size of 8. In the Llama-8B experiments, a batch size of 16 and 2 training epochs are used. Both models are trained with learning rates resulting from minimal hyperparameter tuning (Appendix A). Each task is formatted using a default prompt-completion format (Table 4).

---

[3]https://github.com/allenai/OLMo/tree/main/checkpoints

[4]We also experimented with RedPajama-INCITE-7B (TOGETHER, 2023). After extensive experiments, we found it performed worse than OLMo, given the training data available. Several other models claim to release training checkpoints but have not done so.

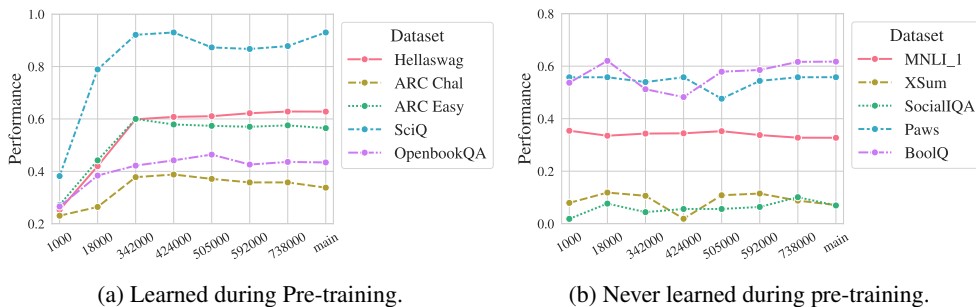

(a) Learned during Pre-training.                    (b) Never learned during pre-training.

Figure 2: Few-shot performance on different pre-training steps.

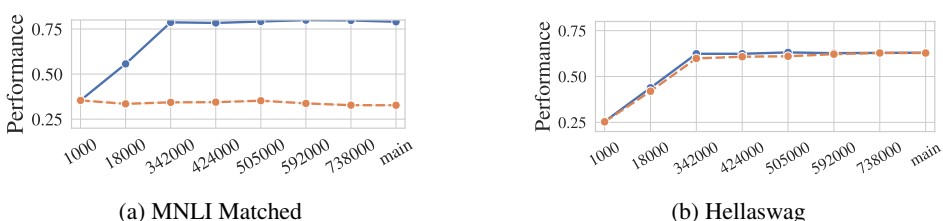

(a) MNLI Matched                              (b) Hellaswag

Figure 3: Example of few-shot performance on different pre-training steps of the models that benefited (3a) and did not benefit from fine-tuning (3b). The **solid blue** line represents the fine-tuned checkpoint, and the **dashed orange** line represents the base checkpoint. The results of all datasets can be found in Figure 10 and Figure 9.

**Instruction Fine-Tuning.** We instruction-tune the model on TÜLU (Ivison et al., 2023), following the decision of Groeneveld et al., 2024. Each model checkpoint is fully fine-tuned for 5 epochs with a batch size of 8 and a learning rate of $2 \times 10^{-6}$.

## 3.3 EVALUATION

The evaluation challenge is to select a representative number of datasets for different types of tasks to test model abilities, recognizing that each dataset requires evaluating each model checkpoint and its fine-tuned counterparts. We also select datasets based on the availability of in-domain and out-of-domain samples.

**Datasets.** The datasets are summarized in Table 1. We evaluate the model with an in-domain test set and one or two out-of-domain test sets for each of the supervised fine-tuning tasks. We conduct experiments on the tasks of *summary generation* (Narayan et al., 2018; Hasan et al., 2021; Hermann et al., 2015), *question generation* (Sap et al., 2019; Xiong et al., 2019; Welbl et al., 2017), *natural language inference* (Williams et al., 2018; Wang et al., 2018; Dagan et al., 2006; Bar Haim et al., 2006; Giampiccolo et al., 2007; Bentivogli et al., 2009), and *paraphrase detection* (Zhang et al., 2019; Wang et al., 2018; Agirre et al., 2007). Each training set is sub-sampled to a size of 6,000 for a fair comparison.

In instruction fine-tuning, we base our downstream evaluation settings on Groeneveld et al. (2024), as OLMo is found to have stable performance on these datasets. The instruction-tuned models are evaluated on ARC (both `arc easy` and `arc challenge`) (Clark et al., 2018), OpenbookQA (Mihaylov et al., 2018), Hellaswag (Zellers et al., 2019), BoolQ (Clark et al., 2019), and SciQ (Welbl et al., 2017).

**Metrics.** We use accuracy (Pedregosa et al., 2011) for classification tasks and ROUGE-L (Lin, 2004) for generation tasks. The maximum amount of newly generated tokens is set to 5 for classification tasks and 60 for generation tasks. Outputs are generated with greedy decoding. For classification tasks, we experiment with both constrained decoding and logit-based predictions. We find the best performance by selecting the label with the highest logit of its first subtoken (Appendix B).

## 4    HOW DOES THE MODEL CHANGE ACROSS PRE-TRAINING?

We begin our evaluation by considering how additional pre-training changes the BASE model. Typically, researchers track the value of the training or held-out loss during training. However, performance improvements on downstream tasks do not always follow the same trend with the loss curves (Groeneveld et al., 2024).

We evaluate the pre-trained checkpoints with few-shot examples, as models without alignment tend to do poorly in a zero-shot context. Four shots are randomly sampled from the datasets, which are selected based on the highest performance shot amount reported in Yang et al. (2024). The model's performance at each pre-training step is reported in Figure 2.

Broadly speaking, our results suggest that all datasets fall into one of two groups. For the first group of datasets (Figure 2a), although the model shows clear improvement during the early stages of pre-training,

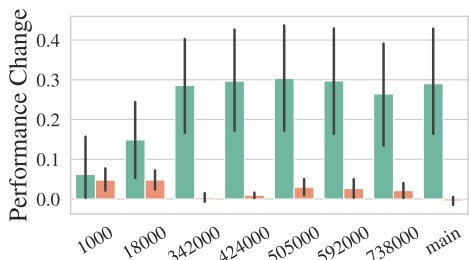

Figure 4: Amount of performance increase brought by fine-tuning between tasks that model can solve in pre-training (**mandarin orange**) and tasks that the model could not solve until fine-tuning (**sage green**). The exact number of mean increases is shown in Appendix I.

performance levels off fairly early on and remains consistent. The dramatic improvements in the early stages of pre-training may result from larger steps in early optimization. We find improvements stop increasing past step 342,000. The second group (Figure 2a) shows tasks that are never learned during pre-training. Performance remains constant throughout the whole pre-training process. These datasets include MNLI, XSum, and BoolQ, and we observe no difference among evaluations with shot sizes of 0, 4, and 7. A natural hypothesis for this finding is potential data contamination in the pre-training data. However, the evaluation datasets are selected based on the popularity of the task and the content of pre-training data. All datasets that experience improvement do not exist in the model's pre-training data (Soldaini et al., 2024), while the more likely leaked datasets (MNLI, XSUM) never gain an improvement during the pre-trining process.

Overall, these results reveal an interesting dichotomy. Some tasks can be learned during pre-training, while others are not. Next, we explore what exactly the model is learning regarding this second group of datasets during pre-training by exploring the fine-tuned models.

## 5    DOES MORE PRE-TRAINING YIELD BETTER FINE-TUNING RESULTS?

Groeneveld et al. (2024) compares OLMo's performance on several tasks before and after fine-tuning the final checkpoint and finds that fine-tuning enables the model to do well on tasks for which the unaligned model does poorly. We observe (§4) that while some datasets improved during pre-training, there is a group of datasets for which a pre-trained model does poorly. Does the model learn anything that helps solve these tasks, and is fine-tuning required to do well on them? Alternatively, does the model learn useful information for these tasks but cannot express it without fine-tuning? In this section, we further explore this dataset dichotomy by examining fine-tuned checkpoints for each of the datasets.

Our results appear in Figure 3 and Figure 4. First, we consider those datasets where the pre-trained models do well (Figure 2a). These datasets do not improve with fine-tuning, suggesting whatever is learned during fine-tuning, which we discuss below, the model already gains the knowledge during pre-training. This effect is observed at all checkpoints; fine-tuning simply does not help.

However, a different story is observed for datasets that are not learned during pre-training. For these, fine-tuning yields significant improvements at every model checkpoint, with Figure 4 showing the magnitude of improvement on these datasets compared to no improvement to the datasets already learned during pre-training. Moreover, earlier checkpoints obtain more substantial gains from fine-tuning than later checkpoints. The benefit of fine-tuning continues to increase until a certain threshold in pre-training steps is reached (approximately 424,000).

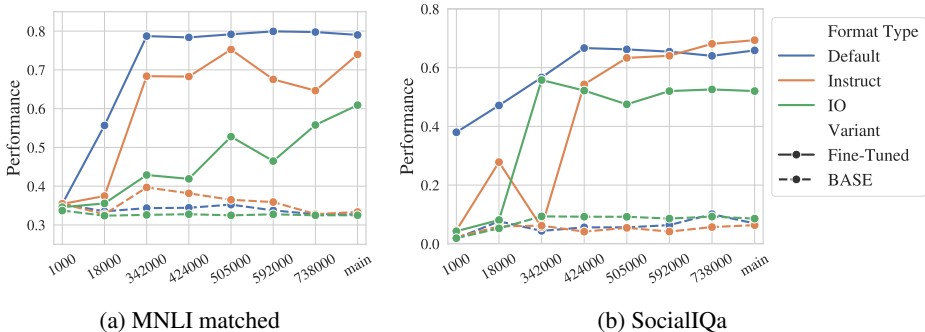

(a) MNLI matched               (b) SocialIQa

Figure 5: Example of model performance with different task formats. The figure of all datasets can be found in Figure 14.

Figure 3 shows representative plots comparing the performance of a pre-trained versus fine-tuned model at different checkpoints for two datasets (full list in Appendix E). For Hellaswag (learned during pre-training), fine-tuning does not benefit the model, even during early checkpoints when the model performs poorly on the task. Nevertheless, for MNLI (not learned during pre-training), fine-tuning dramatically improves the model. Interestingly, later checkpoints achieve better results after fine-tuning, even when the performance of the pre-trained model is unchanged. This suggests that **the model is, in fact, learning important information during pre-training, but it cannot express that information without fine-tuning**.

Our findings suggest that early stopping in pre-training will not be detrimental to downstream fine-tuning performance, and **the benefits of fine-tuning an LLM could exceed the benefits of continued pretraining**, which sheds light on the potential of a cost-effective training paradigm with less pre-training. However, it is difficult to directly identify such a stopping criteria without fine-tuning intermediate checkpoints; the improvement trend is invisible before fine-tuning the checkpoints. Future work may reveal other signals of pre-training behavior that correlate with downstream task performance after fine-tuning. Overall, when resource-intensive pre-trained LLMs are not available, fine-tuning models on models with less pre-training may be a reasonable practical choice for obtaining a high-quality model.

## 6 SUPERVISED FINE-TUNING: WHAT DOES THE MODEL LEARN AND FORGET?

What exactly is the model learning during fine-tuning such that it shows abilities in pre-trained models for some tasks but provides no benefit for other tasks? We analyze the supervised fine-tuning process to understand what is learned and what is forgotten. Specifically, we explore three dimensions: **task format, task transfer, and domain knowledge**.

### 6.1 TASK FORMAT

Several works show that LLMs are extremely sensitive to prompt perturbation in few-shot settings (Sclar et al., 2023; Leidinger et al., 2023; Salinas & Morstatter, 2024; Wahle et al., 2024). We hypothesize that fine-tuning fits the model to a specific task format, resulting in higher performance when the evaluation set matches this format. To test this hypothesis, we vary the task format to either match the training format, use a different format, or rely on instructions. We carefully construct three different prompt formats for the following settings. 1) `Default` is the same format used for training, where we expect the model to benefit from learning the task format; 2) In contrast, `IO` format reflects a common way of performing supervised fine-tuning by incorporating only unprocessed input and output; 3) `Instruct` uses a human-readable instruction template to format the input. Table 4 shows an example of each format. Checkpoint performance on OLMo before and after fine-tuning is shown in Figure 5. The performance of Llama3-8B with different task formats is shown in Figure 6.

In the early pre-training steps, aligning the task format with fine-tuning data plays a crucial role. The model does not yet have enough information to overcome the differences between the training and test formats. However, when fine-tuned on later pre-training checkpoints, the model gradually becomes

more flexible with different task formats, suggesting that model sensitivity to prompt formatting may decrease with more pre-training and a fine-tuning stage. In this view, fine-tuning teaches the model how to format a response for the task, while the instruction provides a directed prior for the model to behave in a certain way. Similarly, on Llama3-8B, we observe that the variation caused by prompt is not affected by fine-tuning and `instruct` preserves its benefits even after fine-tuning. These observations confirm the findings Hewitt et al. (2024) that models retain their instruction-following ability after fine-tuning without instructions.

## 6.2 Task Transfer

Model forgetting occurs when model training on new tasks improves those tasks at the expense of previously trained tasks (Luo et al., 2023; Mehta et al., 2023; Li & Lee, 2024). To understand whether the model will forget a previously known task solution when fine-tuned on a different one, we evaluate model forgetfulness by examining whether the model does worse on some tasks after fine-tuning for other tasks. Specifically, we divide our tasks into two types: classification and generation. We notate the training datasets as $D_T$ and the evaluation

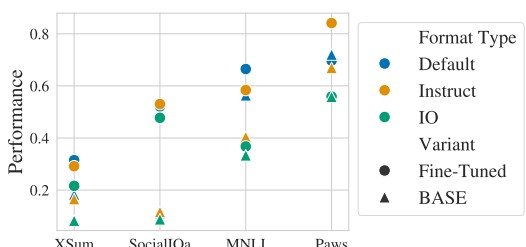

Figure 6: LLAMA3-8B performance with different task format.

datasets as $D_E$. We represent the performance of a pre-trained model (BASE) on checkpoint $i$ as $\text{Perf}^i_{BASE}(d)$ for an evaluation dataset $d \in D_E$, and the performance of the i-th checkpoint fine-tuned on dataset $t \in D_T$ be $\text{Perf}^i_t(d)$. To normalize the effect caused by uneven performance across different datasets, we compute the mean ratio of change (MRC) in performance for each checkpoint as follows.

$$\text{MRC} = \frac{1}{|D_E \setminus \{t\}|} \sum_{\forall d \in D_E, d \neq t} \frac{\text{Perf}^i_t(d) - \text{Perf}^i_{BASE}(d)}{\text{Perf}^i_{BASE}(d)} \tag{1}$$

Models fine-tuned on classification tasks and evaluated on generation tasks decrease on average 61.4% compared to models that are never fine-tuned. In contrast, models fine-tuned on generation tasks can still perform the same as the BASE model on classification tasks, with a 0.3% MRC, which is not statistically significantly different from a 0% change. Our findings on all pre-training checkpoints align with the findings of Yang et al. (2024) on the final checkpoint of LLAMA-7B and our experiments on the final checkpoint of Llama3-8B (Appendix F).

Regardless of the pre-training stage, a model can maintain classification abilities when trained for generation but loses generation abilities when trained for classification. This is perhaps not surprising given that classification tasks can be seen as a subset of generation, while the reverse is not true. The model follows a simplicity bias (Shah et al., 2020) and thus is more likely to memorize simple classification tasks than generation tasks with an exponentially larger search space. Additionally, since we evaluate the classification tasks based on the output logits and the base model performs randomly on the classification tasks, it is much easier for the models to maintain the same performance as the BASE models. Fine-tuning can cause a model to lose abilities when the desired fine-tuning behavior does not support those abilities.

## 6.3 Domain Knowledge

Finally, we explore how a model's generalization ability is affected by fine-tuning by inspecting whether the model forgets the domain knowledge it had before fine-tuning due to learning other abilities. An example of OOD model performance is shown in Figure 7, and the mean ratio of change by datasets is presented in Figure 8 and Figure 15.

The models do not benefit equally from the in-domain fine-tuning: On OLMo checkpoints, all NLI datasets experience a boost when fine-tuning on MNLI, while fine-tuning on Paws is detrimental to other paraphrase detection datasets. This implies that both forgetting and learning are happening: the model learns to perform the task with in-domain knowledge, but it may, in turn, forget information more distant from what is learned in fine-tuning. Questions remain, however, about whether there

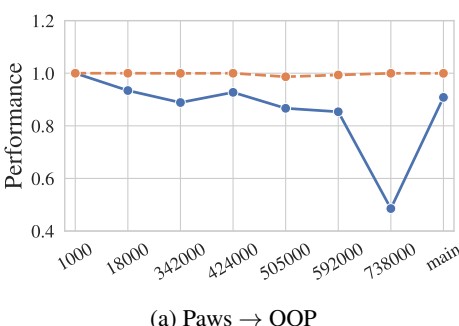
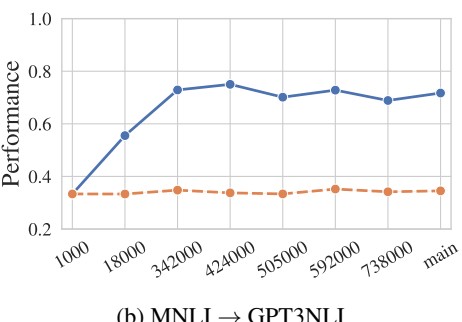

(a) Paws → QQP                    (b) MNLI → GPT3NLI

Figure 7: Example of out-of-domain performance for fine-tuned models. The **solid blue** line represents the fine-tuned checkpoint evaluated on an out-of-domain dataset, and the **dashed orange** line represents the base checkpoint where the model is not fine-tuned. Figure 7a shows an example of fine-tuning hurting OOD performance, while Figure 7b shows an example of fine-tuning boosting OOD performance as pre-traininng proceeds.

are different stages of learning and forgetting during fine-tuning and whether the model picks up different tasks in various stages, which requires further study of fine-tuning dynamics.

Across these three experimental settings, we find that fine-tuning teaches a model how to perform a task, but can sacrifice generalization across domains and tasks. For some datasets learned with pre-training alone, the nature of the task is probably supported by the pre-training objective. For tasks that can only be learned with subsequent fine-tuning, the model may require additional examples or instructions to adapt to different task formats, or the task itself may be inconsistent with the pre-training objective.

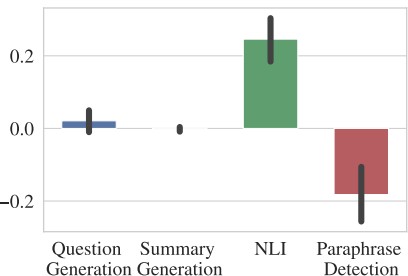

Figure 8: Ratio of out-of-domain performance change for each task, averaged across checkpoints.

## 7 DISCUSSION

Our study fine-tunes model pre-training checkpoints to understand the dynamics of pre-training and fine-tuning on model performance.

*Some datasets can be learned without fine-tuning.* We discover a dichotomy between datasets. Some are learned during model pre-training, while others show no improvements during pre-training. Furthermore, the datasets learned during pre-training do not benefit from fine-tuning. This observation, combined with our study about what is learned during fine-tuning (§6) suggests that some tasks are presented in a manner that aligns with what the model sees during pre-training, and thus fine-tuning provides no additional information. It may be possible to modify tasks to better align with pre-training and thus make them learnable.

*Pre-training can improve models in unseen ways.* Some datasets are not learned during pre-training but benefit significantly from fine-tuning (§4). However, these datasets still benefited from additional pre-training, even though those benefits were not revealed without fine-tuning (§5). The model is learning important information about the task, even though it cannot express that information. Future work may identify ways to detect these improvements during pre-training, which can better guide pre-training choices to produce models that perform better post-fine-tuning. Perhaps there is a way in which information about these tasks can be included in pre-training, allowing the model to better utilize the massive amount of pre-training data. For example, early stopping during pre-training could lead to better utilization of limited training resources if we know when to stop. Our results suggest that early stopping in pre-training and starting fine-tuning is an efficient way of utilizing the resource when the downstream datasets are never picked up by the model during pre-training.

*Fine-tuning teaches task format but leads to forgetting unused abilities.* Our results show that fine-tuning guides the model to understand the format and complete a given task. As this information

diminishes, the model's overall ability improves. However, fine-tuning comes at the expense of other model abilities, such as the capability of solving tasks or domains that are unrelated or weakly related to the fine-tuning task. This insight can be helpful in our understanding of the multitask abilities of LLMs, where certain tasks can introduce conflicts during multi-task training (Mueller et al., 2022).

## 8 LIMITATIONS

While our insights suggest directions for future work, we note important limitations inherent in our experiments. We discuss the weaknesses and limitations in the following section.

**Computing Resource.** Due to computational constraints, we can only conduct experiments on a 1B model and a limited amount of datasets. We supply the final checkpoint of an 8B model to further verify the findings that are shared across checkpoints. The amount of GPU hours spent for each experiment in this study is listed in Table 3.

**Model Size and Variant.** For the analysis with intermediate checkpoints, this study considered a single, relatively small LLM, which may, therefore, conceal the emergent capability brought by larger models (Wei et al., 2022). To combat this, we include the final checkpoint of an 8B model from a different model family. Although this study is conducted on less than a dozen datasets, it still consumes thousands of hours of GPU training time at significant expense. Future work needs to confront these issues on larger models and more datasets.

**Availbility of Pre-training Checkpoints.** This study would benefit significantly from including a broader spectrum of models, but the public pre-training checkpoint releases are limited. Open-source LLMs with intermediate checkpoint release include OLMo (Groeneveld et al., 2024), TinyLLAMA (Zhang et al., 2024), RedPajama-Incite (TOGETHER, 2023), OpenLM (Gururangan et al., 2023a), and Pythia (Biderman et al., 2023). After a series of preliminary experiments, we select these models' best-performing and robust families.

**Analysis Protocol.** Wu et al. (2023) show that the evaluation result may be affected by samples that have been memorized by the model during training instead of revealing the reasoning capability. The only analysis protocol used is the downstream performance of a trained model. More investigation should be done into model internals during pre-training dynamics and how they relate to the effects of fine-tuning.

**Training Paradigm.** Although multiple tuning strategies exist, to create a fair comparison environment where checkpoints receive the same amount of training, models are fine-tuned with a fixed amount of epochs in this work. On different pre-training stages, the model may converge at a different speed. Further study can be done to study the effect of pre-training on different fine-tuning methods or fine-tuning dynamics in different pre-training stages. We only explored the scenario of full-parameter fine-tuning. Whether parameter-efficient fine-tuning or human preference tuning will lead to a different conclusion also remains an open question.

**Randomness.** In this study, we only assess uncertainty with Bootstrap during evaluation. However, uncertainty may emerge during training, which poses optimizer initialization and data ordering, the study of which requires an extensive amount of computing resources.

## 9 CONCLUSION

We explore the relationship between fine-tuning and pre-training LLMs through fine-tuning multiple pre-training checkpoints of large language models. Our results on 18 datasets provide insights into LLM training. We find that continual pre-training improves the model in a latent way that is only observable after fine-tuning. The model may excel at some tasks without fine-tuning. However, the model can rapidly learn datasets that it does not demonstrate capabilities during pre-training with only a small amount of supervision. In the meantime, we identify the aspects that LLM learns and forgets during supervised fine-tuning: task format, task solution, and domain knowledge. Overall, our study highlights the value of analyzing language model training dynamics, in line with Chen et al. (2023). We would like to advocate for the release of pre-training checkpoints, the release of which will enable the broader research community to conduct more in-depth future research.

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

| Dataset | Model | Free | Constrained | TokenProb |
|---------|-------|------|-------------|-----------|
| MNLI | Fine-tuned | 0.786 | 0.791 | 0.792 |
|      | BASE | 0.0 | 0.0 | 0.327 |
| RTE | Fine-tuned | 0.658 | 0.662 | 0.66 |
|     | BASE | 0.0 | 0.0 | 0.241 |
| Paws | Fine-tuned | 0.871 | 0.878 | 0.878 |
|      | BASE | 0.0 | 0.0 | 0.558 |
| STS-B | Fine-tuned | 0.775 | 0.741 | 0.744 |
|       | BASE | 0.0 | 0.0 | 0.964 |

Table 2: Performance of final checkpoint with different prediction generation method.

## A  HYPERPARAMETER TUNING

For both supervised fine-tuning and instruction tuning, we pre-set the effective batch size to 8, and tune the learning rate within $\{2 \times 10^{-5}, 2 \times 10^{-6}, 2 \times 10^{-7}\}$. OLMo-1B is fine-tuned for 3 epochs on the supervised fine-tuning tasks and 5 epochs on Tulu for instruction tuning. Llama3-8B is fine-tuned for 2 epochs with a learning rate of $5 \times 10^{-6}$, with learning rate selected from $\{5 \times 10^{-5}, 5 \times 10^{-6}, 5 \times 10^{-7}\}$. In both settings, we adopt an AdamW optimizer with a linear learning rate scheduler. The optimizer is warmed up for the first $3\%$ of the training time.

## B  PREDICTION GENERATION METHOD

For classification tasks, we examine three different prediction generation methods: Free Generation (Free), Constrained Generation (Constrained), and Token Probability (TokenProb), the results are shown in Table 2. In Constrained, we force the output to include at least one label in the acceptable label set. In TokenProb, we compare the logits of acceptable labels and select the label with the highest score as the final output. This ablation study is conducted only on the BASE and fine-tuned versions of the final checkpoint of the pre-trained model. We find that, although prediction generation methods have less effect on the evaluation result of a fine-tuned model, BASE variants suffer much more from not knowing the desired output. Therefore, we proceed with all the classification experiments with TokenProb.

### B.1  LABEL AND TOKENIZATIONS

Depending on the tokenizer variant, the label text may be tokenized differently, leading to evaluation unreliability. For example, in paraphrase detection, the model could assign probability on both "yes" and " yes" (the same label with a prefix space). This behavior is reported and explored in various related work (Sun et al., 2023; Batsuren et al., 2024; Singh & Strouse, 2024). In this study, we leniently regard all individual tokens that contain the whole label or part of the label along with some special characters that do not affect the semantics as an acceptable target label.

## C  TASK FORMAT

We adopt the task format from (Yang et al., 2024), with an additional task format of input-output. How each dataset is formated can be found in Table 4.

## D  GPU HOURS PER-EXPERIMENT

We show a table of GPU hours spent for each experiment in Table 3. The total number of GPU hours spent on this project is approximately 1067 A100 hours. We lose track of the GPU hours spent on preliminary experiments, so a lower-bound estimation is reported.

| Prelinminary Experiments | |
| --- | --- |
| **Description** | **GPU Hours** |
| Instruction Tuning on LIMA, TULU, and NaturalInstructions | $\sim$300 |
| Model Performance Verification, Dataset Selection | 120 |
| **Instruction Tuning** | |
| Instruction Tuning | 360 |
| Evaluation | 10 |
| Total | **370** |

| **Fine-Tuning** | | | | |
| --- | --- | --- | --- | --- |
| | **XSum** | **SocialIQa** | **MNLI** | **Paws** |
| **Training** | 12 | 6 | 4.6 | 5.3 |
| **Evaluation** | 8 | 5.3 | 3 | 2 |
| **OOD Evaluation** | 96 | 32 | 11 | 25.6 |
| **CrossTask Evauation** | 5.2 | 6.5 | 7.7 | 8.15 |
| **Task Format Evaluation** | 16 | 12.8 | 6 | 4 |
| **Total** | 137.2 + 62.6 + 32.3 + 45 = 277.1 | | | |

Table 3: GPU hours for each experiment. The total amount of GPU hours spent in this project is approximately 1067 A100 hours.

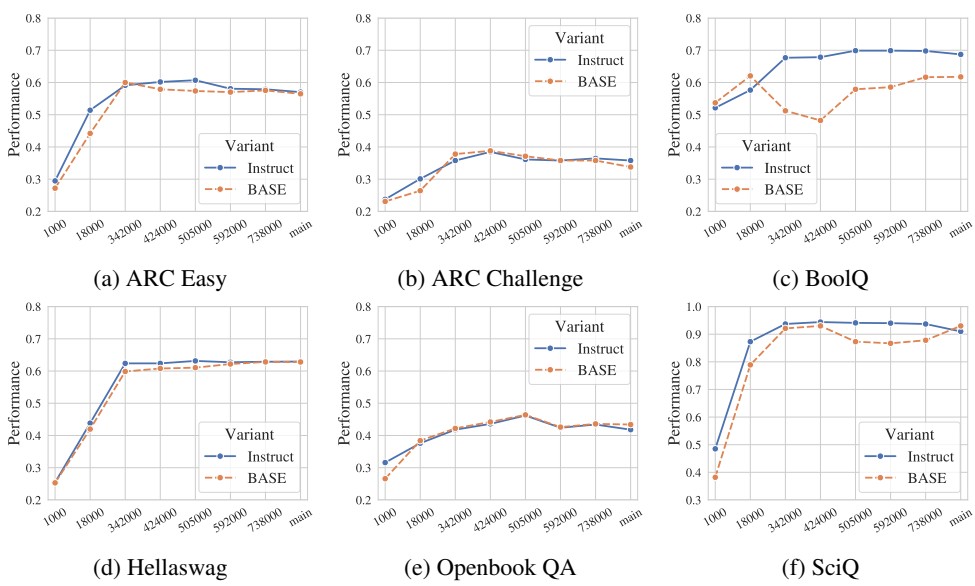

(a) ARC Easy    (b) ARC Challenge    (c) BoolQ

(d) Hellaswag    (e) Openbook QA    (f) SciQ

Figure 9: Model performance after instruction tuning on each pre-training step.

# E    PER-DATASET FIGURES

We show the model performance on each dataset after supervised fine-tuning and instruction tuning correspondingly in Figure 10 and Figure 9. The datasets that already show improvement during pre-training do not benefit from fine-tuning, while performance improves drastically on the datasets that the model has never learned during pre-training.

**Out-of-domain Generalization**  The out-of-domain performance for each dataset with respect to pre-training steps is shown in Figure 11. Overall, the model generalizes well after fine-tuning on NLI tasks, while its performance deteriorates when evaluated on out-of-domain paraphrase detection tasks.

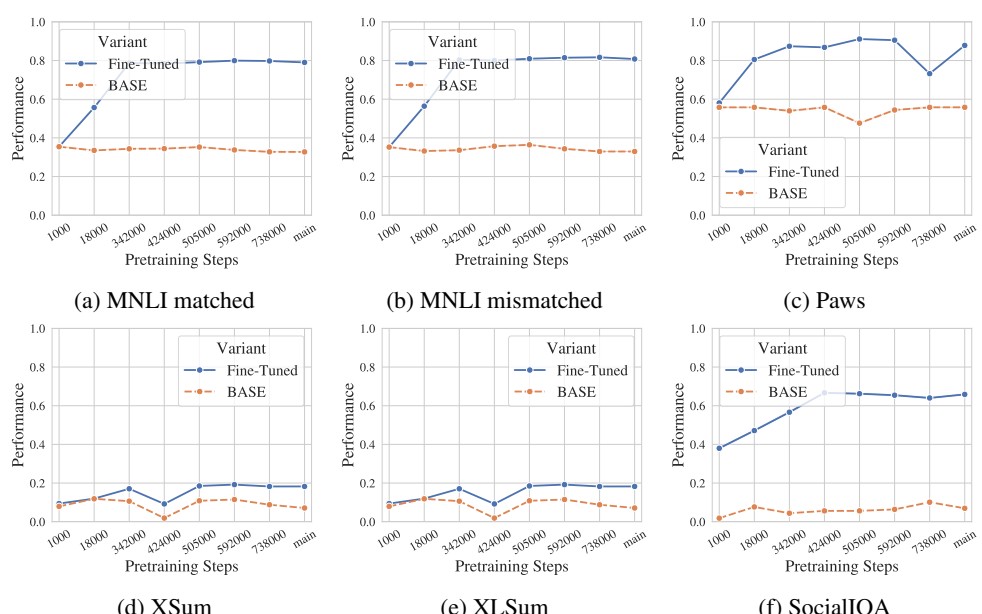

Figure 10: Model performance after supervised fine-tuning on each pre-training step.

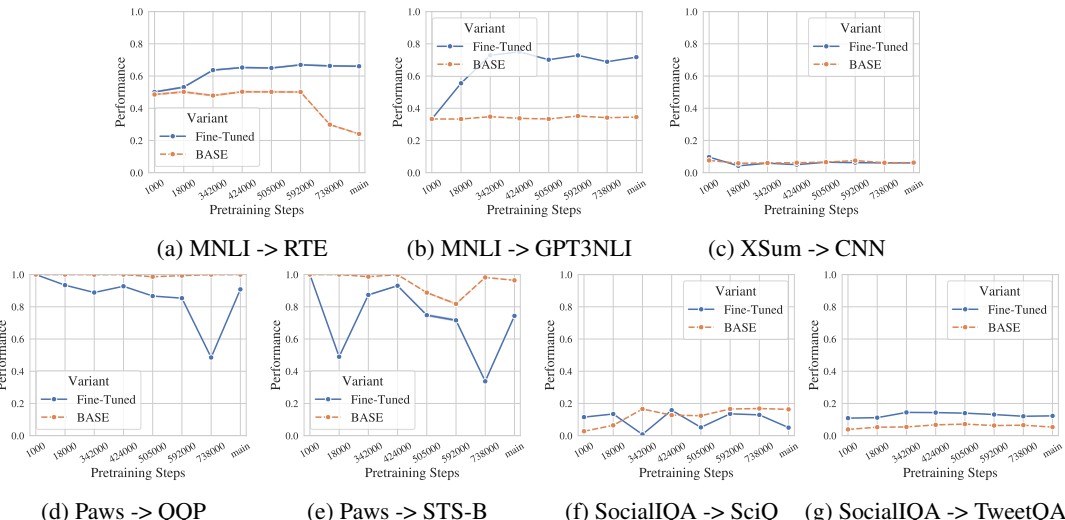

Figure 11: Out-of-domain performance after supervised fine-tuning on each pre-training step.

**Cross-task Generalization** The cross-task performance for each dataset with respect to pre-training steps is shown in Figure 12 and Figure 13.

**Task-Format** The performance of models on evaluation sets formatted with different prompt formatting methods is shown in Figure 14.

# F  LLAMA3-8B RESULTS

To provide more evidence of the findings on a different model architecture and size, we conduct some experiments on the final checkpoint of Llama3-8B.

**Task Transfer** Similar to our findings with OLMo, Llama3-8B fine-tuned on classification tasks and evaluated on generation tasks decreases on average 61.0% compared to models that are never

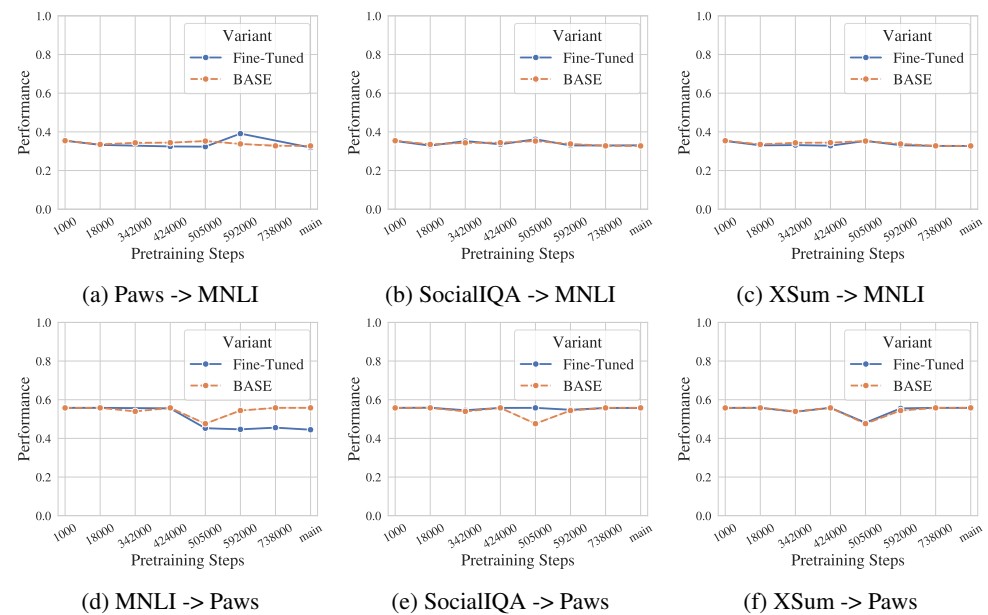

Figure 12: Cross-task performance after supervised fine-tuning on each pre-training step. The model is fine-tuned on a classification task and evaluated on a generation task or a classification task with a different label set.

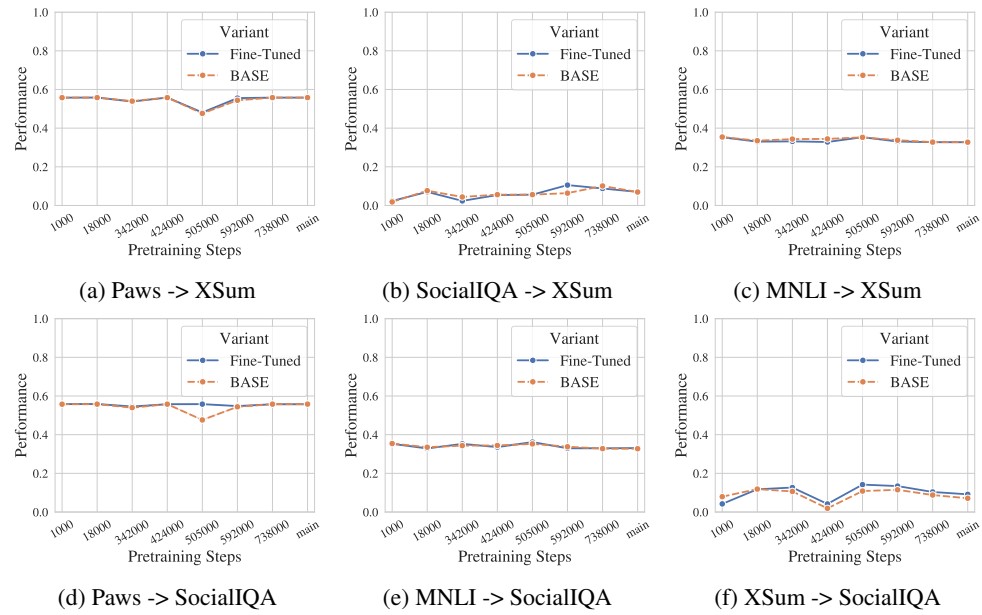

Figure 13: Cross-task performance after supervised fine-tuning on each pre-training step. The model is fine-tuned on a generation task and evaluated on a classification task.

fine-tuned. In contrast, models fine-tuned on generation tasks perform similarly as the BASE model on classification tasks, with a 10.6% MRC.

**Domain Knowledge** The ratio of out-of-domain performance change by task is shown in Figure 15. Overall, we observe that Llama and OLMo experience benefits with different tasks after fine-tuning, but both model shows an inconsistent change across tasks.

| Task | Default Prompt | Instruction Prompt | IO Prompt | Expected Output |
|---|---|---|---|---|
| **Summary Generation** | ### Input: {document} 
 ### Summary: | Please read the following text: {document} 
 Provide a summary: | {document} | {summary} |
| **Question Generation** | ### Input: {context} 
 ### Answer: {answer} 
 ### Question: | Given the context: {context} 
 And the answer: {answer} 
 Generate a suitable question: | {context} 
 {answer} | {question} |
| **Natural Language Inference** | ### Input_1: {premise} 
 ### Input_2: {hypothesis} 
 ### Inference: | Consider the following texts: Text 1: {premise} 
 Text 2: {hypothesis} The relation is | {premise} 
 {hypothesis} | {label} |
| **Paraphrase Detection** | ### Input_1: {sentence1} 
 ### Input_2: {sentence2} 
 ### Paraphrase Classification: | Let's compare the two sentences: 
 Sentence_1: {sentence1} 
 Sentence_2: {sentence2} Are they paraphrasing?: | {sentence1} 
 {sentence2} | {label} |

Table 4: Formatting of the prompts

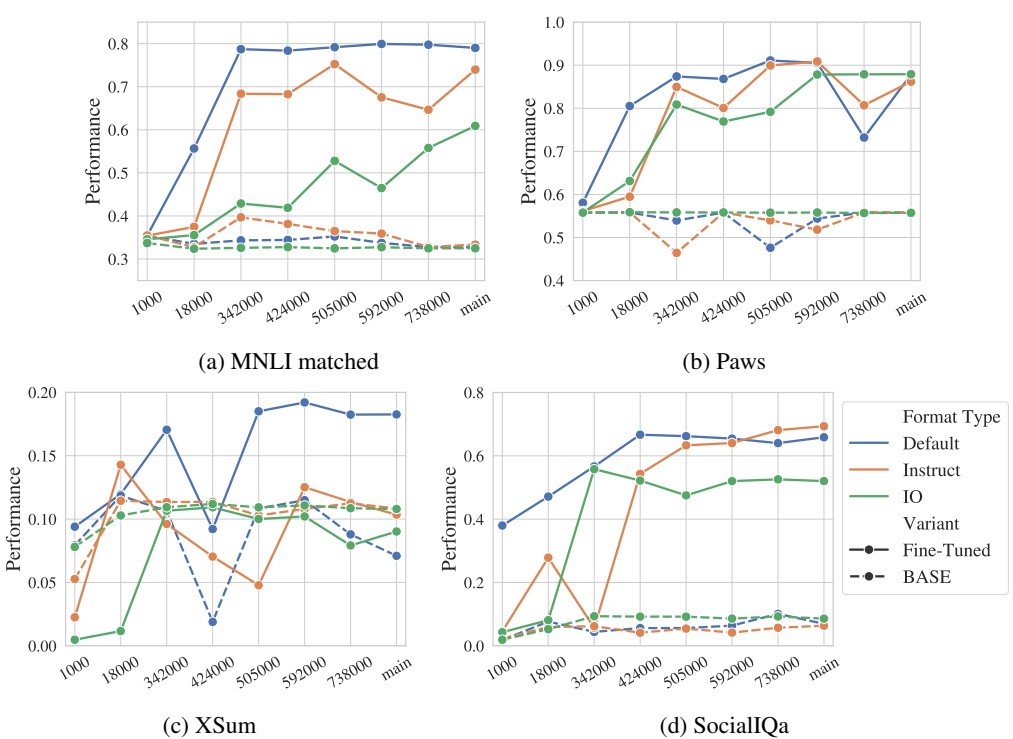

(a) MNLI matched

(b) Paws

(c) XSum

(d) SocialIQa

Figure 14: Model performance with different task formats.

# G  LICENSE OF ARTIFACTS

We include the license of artifacts used in this paper in Table 6

# H  FULL PERFORMANCE TABLE

Due to the availability of space and the amount of fine-tuned checkpoints, we omit displaying all exact metric values in the paper. The performance of each fine-tuned variant on each dataset can be found in the csv file under directory results in the code base.

# I  PERFORMANCE DIFFERENCE NUMBERS

The average performance change before and after fine-tuning for each checkpoint is shown in Table 5. The data in this table is used to create Figure 4.

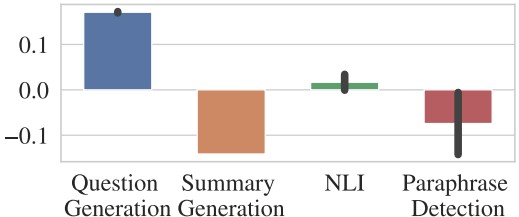

Figure 15: Ratio of out-of-domain performance change for each task on the final checkpoint of LLAMA3-8B.

| Checkpoint | Learned in Pre-train | Learned in Fine-Tune |
|---|---|---|
| 1000 | 0.048 | 0.062 |
| 18000 | 0.048 | 0.149 |
| 342000 | 0.004 | 0.286 |
| 424000 | 0.01 | 0.297 |
| 505000 | 0.03 | 0.304 |
| 592000 | 0.027 | 0.297 |
| 738000 | 0.021 | 0.264 |
| main | -0.005 | 0.290 |

Table 5: Average performance change before and after fine-tuning for each checkpoint (Perf(Fine-tuned) - Perf(BASE)). The group that is never learned during pre-training is picked up by the model during fine-tuning.

## J GENERALIZATION TAXONOMY

Following the generalization taxonomy in Hupkes et al. (2023), the evaluation card is included in Table J.

| Motivation | | | | | |
|---|---|---|---|---|---|
| *Practical* □ △ | *Cognitive* | | *Intrinsic* | | *Fairness* |
| **Generalisation type** | | | | | |
| *Compositional* | *Structural* | *Cross Task* △ | *Cross Language* | *Cross Domain* □ | *Robustness* |
| **Shift type** | | | | | |
| *Covariate* □ | *Label* △ | | *Full* | | *Assumed* |
| **Shift source** | | | | | |
| *Naturally occuring* □ △ | *Partitioned natural* | | *Generated shift* | | *Fully generated* |
| **Shift locus** | | | | | |
| *Train–test* | *Finetune train–test* □ △ | | *Pretrain–train* | | *Pretrain–test* |

| Name | License | Name | License |
|------|---------|------|---------|
| OLMo-1b | Apache 2.0 | SocialIQa | CC-BY |
| TULU | ODC-BY | CNN/DailyMail | Apache 2.0 |
| ARC | CC BY-SA | TweetQA | CC BY-SA-4.0 |
| OpenbookQA | Apache 2.0 | MNLI | CC-BY-3.0 |
| Hellaswag | MIT | GPT3NLI | MIT |
| BoolQ | Apache 2.0 | RTE | N/A |
| SciQ | CC-BY-NC-3.0 | Paws | Free |
| XSum | MIT | QQP | Non-Commercial |
| XLSum | CC-BY-NC-SA 4.0 | STS-B | Other |

Table 6: License of artifacts used in this paper.

