# OpenReview forum: "Amuro and Char: Analyzing the Relationship between Pre-Training and Fine-Tuning of Large Language Models"
_ICLR.cc/2025/Conference — ICLR 2025 Conference Withdrawn Submission_

### Official Review · Reviewer_VUcu · 2024-10-30

**Soundness:** 2
**Presentation:** 2
**Contribution:** 2
**Rating:** 3
**Confidence:** 5

**Summary:**

The paper explores the relationship between fine-tuning and pre-training LLMs through fine-tuning multiple pre-training checkpoints of large language models.

There are some findings based on experimental results:
- The pre-trained model may excel at some tasks without fine-tuning.
- Continual pre-training improves the model in a latent way that is only observable after fine-tuning.
- The fine-tuned model may forget some unused abilities.
- The fine-tuned model exhibits high sensitivity to evaluation prompts, but this sensitivity can be alleviated through more pre-training

**Strengths:**

- Exploring the relationship between pre-training and fine-tuning is a valuable direction with significant implications for improving training efficiency and downstream task performance.
- The paper conducts a series of experimental analyses and summarizes some conclusions, which have some guiding significance for researchers who are new to the field.

**Weaknesses:**

- The conclusion drawn from the paper is relatively superficial and has been discussed in many previous works or some industry consensus, which does not meet the bar of an ICLR paper.
- The paper lacks some deeper insights into analyzing the parameter changes or loss changes during the pre-training or fine-tuning stages, which would provide theoretical support for the observed experimental phenomena.
- The paper's layout is somewhat chaotic, with some figures/tables and related text not on the same page, which poses a significant obstacle to reading.

**Questions:**

- In Section 5, the author claims that "the benefits of fine-tuning an LLM could exceed the benefits of continued pretraining", but in Section 7, the author also claims that "pre-training can improve models in unseen ways". These two viewpoints seem contradictory.
- During the fine-tuning process, the paper conducts experiments on different specific tasks. What if it is in a general setting (such as AlpacaEval, MT-Bench), would the conclusions be different?

---

> ### Author Response · Authors · 2024-11-19
>
> **“The conclusion drawn from the paper is relatively superficial… ”**
>
> While we acknowledge the limitations of our study, we believe it offers valuable insights into a relatively unexplored area of the training process. Our findings highlight an important starting point that can inspire and guide future work. We hope this study demonstrates the value of pre-training checkpoints and encourages model builders to make them more widely available.
>
> To the best of our knowledge, no prior work has explored pre-training checkpoint experiments. Given the lack of alternative models to evaluate and the substantial resources we dedicated—over 1100 A100 GPU hours—we believe this work represents a significant step forward in this domain. We hope it underscores the feasibility of such research for the academic community, even within resource constraints.
>
> **“... and has been discussed in many previous works or some industry consensus”**
>
> We kindly request clarification or references to the specific previous work that the reviewer believes our conclusions overlap with. Without knowing the specific work the reviewer is referring to, it is challenging to address this concern or highlight the distinctions and contributions of our study effectively.
>
> **“The paper lacks some deeper insights into analyzing the parameter changes or loss changes during the pre-training or fine-tuning stages, which would provide theoretical support for the observed experimental phenomena.”**
>
> We believe our study still offers valuable insights. Throughout these analyses, we provide a meaningful examination of the training process that is often overlooked due to the current nature of training large language models. Moreover, our findings serve as an important starting point that can motivate and guide future work
> Loss changes have already been reported by the authors who released the models we used [1, 2]. We think our findings would complement the existing work that only reports loss changes.
>
> **“The paper's layout is somewhat chaotic, with some figures/tables and related text not on the same page, which poses a significant obstacle to reading.”**
>
> We believe these are easily addressable issues. If the reviewer can kindly point to the figures/tables mentioned, we can promptly make edits and upload a revision.
>
> **“In Section 5, the author claims that "the benefits of fine-tuning an LLM could exceed the benefits of continued pretraining", but in Section 7, the author also claims that "pre-training can improve models in unseen ways". These two viewpoints seem contradictory.”**
>
> These two findings are indeed not contradictory. Although pre-training can improve models in unseen ways, the improvement is not forever, there are diminishing returns. When the benefits of pre-training set plateaus, which can be identified by observing the datasets that show improvement in the early stage of training, it suggests that fine-tuning an LLM would exceed the benefits of continual pre-training.
>
> **“During the fine-tuning process, the paper conducts experiments on different specific tasks. What if it is in a general setting (such as AlpacaEval, MT-Bench), would the conclusions be different?”**
>
> Even though our datasets seem simple, the model does poorly on them during pre-training. Furthermore, we would like to clarify that our study's primary focus is the impact of supervised fine-tuning, not instruction following. The datasets (e.g., MT-Bench, Alpaca-Eval, Arena-Hard) are specifically designed with instructions, which are orthogonal to our core research questions. Instruction-heavy benchmarks introduce an additional confounding factor—namely, the model's instruction-following ability—rather than the core task-solving abilities we aim to study.
> That said, we agree that exploring the intersection of fine-tuning and instruction-following ability is an interesting direction for future work, and we hope our current findings can serve as a foundation for such analyses.
>
> [1] The Llama 3 Herd of Models
>
> [2] OLMo: Accelerating the Science of Language Models

---

### Official Review · Reviewer_GgqV · 2024-10-31

**Soundness:** 2
**Presentation:** 3
**Contribution:** 3
**Rating:** 5
**Confidence:** 4

**Summary:**

This paper investigate the relationship between pre-training and fine-tuning by fine-tuning multiple intermediate pre-trained model checkpoints to understand how models develop as they train. The authors conduct experiments on 18 datasets and give following insights into LLM training based on the result:
(1) continued pretraining can improve a model in ways that are only revealed after fine-tuning;
(2) tasks for which the model already performs well during pre-training benefit much less from fine-tuning than those where the model does not demonstrate capabilities;
(3) although supervised fine-tuning can improve performance on in-distribution tasks, it can also cause the model to forget domain knowledge or tasks that it was previously capable of solving;
(4) fine-tuned models show high sensitivity to evaluation prompts, but this sensitivity can be alleviated by more pre-training.

**Strengths:**

**(1) The problem that this paper seeks to respond is important and valuable.** E.g., How do pretraining and fine-tuning interact to produce the resulting model? Does more pre-training hinder better fine-tuning results? What does the model learn and forget during pre-training and fine-tuning? These questions are straightforward and valuable.

**(2) This paper is well written.** The author clearly clarifies the problem that each part tries to address, making it easy to understand.

**(3) The author clearly states the limitations of their work.** It is always good to see the authors states the limitations as it makes the paper more rigorous.

**Weaknesses:**

**(1) The experiments are insufficient.** To explore the relationship between pretraining and fine-tuning, it is necessary to ensure the generalizability of the conclusions. Verifying only one language model (OLMo-1B) is insufficient to provide convincing conclusions. I believe the author needs to validate their conclusions on more LLMs.

**(2) Some of the conclusions are not rigorous.** e.g. line 300-303, the authors state that "some tasks can be learned during pre-training, while others are not." This may be because the pretraining data possibly includes data from similar types of tasks (not necessarily contamination), whereas tasks that cannot be learned during pretraining (such as MNLI, XSum, and BoolQ) do not have such similar task data included in their pretraining datasets. In such case, the conclusion become completely meaningless. I suggest the author carefully examine the types of tasks included in the pretraining dataset before drawing conclusions.

**(3) Some insights are uninspired with limited practical guidance value.** E.g., the authors suggest that early stopping in pre-training and starting fine-tuning is an efficient way of utilizing the resource when the downstream datasets are never picked up by the model during pre-training. However, the practical issue is that if we want to train a specialized model, we should directly choose a well-pretrained model. We typically don't aim to start from the pretraining phase again. The primary purpose of pretraining is to equip the model with stronger foundational capabilities, providing a solid base for better specialization through further SFT.

**Questions:**

See weaknesses.

typos:
line 299: pre-trining->pre-training

---

> ### Author Response · Authors · 2024-11-19
>
> **“ Verifying only one language model (OLMo-1B) is insufficient to provide convincing conclusions.”**
>
> Our study includes two models, OLMo-1b and Llama3-8B, and results on both models reach the same conclusions. Experiments that required pre-training checkpoints only include OLMo since Llama did not release checkpoints. We conducted an exhaustive search for pre-training checkpoints, including contacting several model authors. We are aware of only a few other models with checkpoints, which all have issues. 1) TinyLlama fixed a token problem mid-pre-training, which changed model behavior during pre-training. 2) RedPjama, which we experimented with extensively but performed poorly across all of our fine-tuning experiments. 3) Baichuan2, which is multi-lingual (introduces other issues) and is relatively unknown. 4) LLM360, which has a staged pre-training process that deviates from most other models.
>
> While we acknowledge the limitations of our study, we believe it offers valuable insights into a relatively unexplored area of the training process. Our findings highlight an important starting point that can inspire and guide future work. We hope this study demonstrates the value of pre-training checkpoints and encourages model builders to make them more widely available.
>
> To the best of our knowledge, no prior work has explored pre-training checkpoint experiments. Given the lack of alternative models to evaluate and the substantial resources we dedicated—over 1100 A100 GPU hours—we believe this work represents a significant step forward in this domain. We hope it underscores the feasibility of such research for the academic community, even within resource constraints.
>
> **“However, the practical issue is that if we want to train a specialized model, we should directly choose a well-pretrained model. We typically don't aim to start from the pretraining phase again.”**
>
> An LLM has to be pre-trained from scratch in order to become a “well-pretrained model”. By studying the effect of different amount of pre-training on resulting fine-tuning performance, we are hoping to provide insights on how pre-trained should be done.
> In addition, licensing restrictions on modern LLMs often prevent the selection of pre-trained models in some use cases. In such scenarios, companies frequently pre-train their own LLMs, making the insights from our study on fine-tuning and model optimization highly relevant for real-world applications.

---

> > ### Comment · Reviewer_GgqV · 2024-11-19
> > **Response to authors reply**
> >
> > Thanks for your detailed response. Your response has alleviated my concerns regarding the experiments conducted on a limited number of models. I also highly appreciate the authors for undertaking this research despite the scarce availability of pre-trained model checkpoint resources, which makes a significant step forward in this field. Additionally, I agree with the application scenario described by the authors, namely that companies frequently pre-train their own LLMs for special industry scenarios, and I suggest the authors emphasize this in the paper. However, considering that the authors still not reply weakness(2) I mentioned before, I still cannot recommend accepting this paper at this time. I have adjusted the corresponding scores based on the above.

---

> > > ### Author Response · Authors · 2024-11-22
> > >
> > > Thank you for your thoughtful feedback! We acknowledge the concern about task leakage. Most work in this space, including ours, employs standard checks for data leakage—ensuring that test examples themselves do not appear in the pretraining corpus. However, there are no rigorous methods available to evaluate the broader influence of task similarity within the pretraining dataset. This limitation is indeed a weakness of our study, but it is one that is shared widely across the field. Because this issue is a systemic challenge in the evaluation of LLMs, we believe it would be reasonable to add the discussion of such an issue in a revised draft.

---

### Official Review · Reviewer_Rcfe · 2024-11-02

**Soundness:** 2
**Presentation:** 2
**Contribution:** 2
**Rating:** 5
**Confidence:** 3

**Summary:**

This work analyzes the relationship between Pre-Training and Fine-Tuning of Large Language Models. The authors conduct experiments on multiple intermediate pre-trained checkpoints to analyze how models develop as they train. Through experimental results, they find i) continual pretraining improves the model in a latent way that manifests after fine-tuning; ii) fine-tuning most benefits datasets where the model does not show capability during pre-training; iii) although the model benefits significantly through supervised fine-tuning, it may forget previously known domain knowledge and tasks not seen during fine-tuning; iv) the model exhibits high sensitivity to evaluation prompts after supervised fine-tuning, but this sensitivity can be alleviated through more pre-training

**Strengths:**

(1)	This work explores an interesting topic in LLMs by investigate the relationship between pre-training and fine-tuning.

(2)	The authors conduct some experiments provide some observations in LLM training.

**Weaknesses:**

(1)	There are some observations that are relatively easy to obtain (e.g., although the model benefits significantly through supervised finetuning, it may forget previously known domain knowledge and tasks not seen
during fine-tuning), which have limited impact on the literature.

(2)	The authors should provide a related work section to summarize the difference between this work and previous related studies.

(3)	The model backbone selected in this work is limited (only OLMo model). Have you tried other open-source models (e.g., OpenELM).

**Questions:**

see Weaknesses

---

> ### Author Response · Authors · 2024-11-19
>
> **“The authors should provide a related work section to summarize the difference between this work and previous related studies.”**
>
> Section 2 Background (L88-L147) includes the related work section, in which we discuss a survey of previous work and its relationship with some of the most relevant prior works [1, 2].
>
> **“The model backbone selected in this work is limited (only OLMo model). Have you tried other open-source models (e.g., OpenELM).”**
>
> Our study includes two models, OLMo-1b and Llama3-8B, and results on both models reach the same conclusions. Experiments that required pre-training checkpoints only include OLMo since Llama did not release checkpoints. We conducted an exhaustive search for pre-training checkpoints, including contacting several model authors. We are aware of only a few other models with checkpoints, which all have issues. 1) TinyLlama fixed a token problem mid-pre-training, which changed model behavior during pre-training. 2) RedPjama, which we experimented with extensively but performed poorly across all of our fine-tuning experiments. 3) Baichuan2, which is multi-lingual (introduces other issues) and is relatively unknown. 4) LLM360, which has a staged pre-training process that deviates from most other models.
>
> While we acknowledge the limitations of our study, we believe it offers valuable insights into a relatively unexplored area of the training process. Our findings highlight an important starting point that can inspire and guide future work. We hope this study demonstrates the value of pre-training checkpoints and encourages model builders to make them more widely available.
>
> To the best of our knowledge, no prior work has explored pre-training checkpoint experiments. Given the lack of alternative models to evaluate and the substantial resources we dedicated—over 1100 A100 GPU hours—we believe this work represents a significant step forward in this domain. We hope it underscores the feasibility of such research for the academic community, even within resource constraints.
>
> [1] Sudden Drops in the Loss: Syntax Acquisition, Phase Transitions, and Simplicity Bias in MLMs
>
> [2] Scan and Snap: Understanding Training Dynamics and Token Composition in 1-layer Transformer

---

> > ### Comment · Reviewer_Rcfe · 2024-11-26
> > **Thanks for your response**
> >
> > Thanks for the response. I will keep my score unchanged. Experiments on more models should be considered.

---

### Official Review · Reviewer_UJu9 · 2024-11-04

**Soundness:** 2
**Presentation:** 3
**Contribution:** 2
**Rating:** 3
**Confidence:** 4

**Summary:**

This paper investigates the dynamics of capability acquisition in large language models (LLMs) and provides emprical analyses that reveal the contribution of the pre-training and fine-tuning stages to downstream capabilities. Multiple intermediate pre-training checkpoints were fine-tuned and evaluated, leading to four main findings:
1）the pre-training stage can enhance the performance of the fine-tuned model, even when such improvements are not apparent in the pre-trained model itself;
2）fine-tuning is more beneficial for tasks that have not been learned during the pre-training stage;
3）a model fine-tuned for specific tasks may forget knowledge and capabilities in other domains;
4）fine-tuned models show high sensitivity to evaluation prompts, but this sensitivity can be alleviated by more pre-training.

**Strengths:**

This paper analyze the downstream performance of intermediate pre-training checkpoints and the corresponding fine-tuned models, and draws some insights that can help in developing more efficient and effective LLMs.

**Weaknesses:**

1) The experiment employed only a single base model, which limits the generalization of the empirical findings. In addition to the five candidate models mentioned by the authors, Baichuan2-7B may also be considered a candidate that has released intermediate checkpoints. https://huggingface.co/baichuan-inc/Baichuan2-7B-Intermediate-Checkpoints
2) The parameters of the base model used in this paper amount to 1 billion, which does not include widely used model sizes of LLMs, such as 7 billiion.
3) The num of tasks for supervised fine-tuning is relatively limited, with only 4 tasks, including summary generation, question generation, natural language inference and paraphrase detection. This limits the generalization of the results.
4) The conclusions derived from the empirical analysis largely align with the established perspectives within this field, providing limited novelty.
5) There are no promising experiments demonstrating how these findings can inform the developing of LLMs.

**Questions:**

none

---

> ### Author Response · Authors · 2024-11-19
>
> **“The experiment employed only a single base model, which limits the generalization of the empirical findings.”**
>
> **“The parameters of the base model used in this paper amount to 1 billion, which does not include widely used model sizes of LLMs, such as 7 billion.”**
>
> **“In addition to the five candidate models mentioned by the authors, Baichuan2-7B may also be considered a candidate that has released intermediate checkpoints. https://huggingface.co/baichuan-inc/Baichuan2-7B-Intermediate-Checkpoints”**
>
> Our study includes two models, OLMo-1b and Llama3-8B, and results on both models reach the same conclusions. Experiments that required pre-training checkpoints only include OLMo since Llama did not release checkpoints. We conducted an exhaustive search for pre-training checkpoints, including contacting several model authors. We are aware of only a few other models with checkpoints, which all have issues. 1) TinyLlama fixed a token problem mid-pre-training, which changed model behavior during pre-training. 2) RedPjama, which we experimented with extensively but performed poorly across all of our fine-tuning experiments. 3) Baichuan2, which is multi-lingual (introduces other issues) and is relatively unknown. 4) LLM360, which has a staged pre-training process that deviates from most other models.
>
> While we acknowledge the limitations of our study, we believe it offers valuable insights into a relatively unexplored area of the training process. Our findings highlight an important starting point that can inspire and guide future work. We hope this study demonstrates the value of pre-training checkpoints and encourages model builders to make them more widely available.
>
> To the best of our knowledge, no prior work has explored pre-training checkpoint experiments. Given the lack of alternative models to evaluate and the substantial resources we dedicated—over 1100 A100 GPU hours—we believe this work represents a significant step forward in this domain. We hope it underscores the feasibility of such research for the academic community, even within resource constraints.
>
> **“The conclusions derived from the empirical analysis largely align with the established perspectives within this field, providing limited novelty.”**
> To the best of our knowledge, this is the first paper to study training dynamics using pre-training checkpoints. Please let us know which papers have conducted a similar analysis and where our conclusions have been previously published in the literature.
>
> **“There are no promising experiments demonstrating how these findings can inform the developing of LLMs.”**
>
> We believe that the question reviewer Reviewer S7Wz might be helpful in demonstrating the practical suggestions, so we pasted it here.
>
>
> **“Can you elaborate on potential signals or metrics during pre-training that could indicate an optimal point to stop pre-training and begin fine-tuning?”**
> Empirically, a practical approach is to use a set of validation datasets (for example, datasets in the first exp section) that have been examined to improve throughout pre-training. Those datasets do not require fine-tuning, but they can approximately entail the time when pre-training is sufficient.
> Once the performance on these validation sets plateaus or stops improving, it generally signals a diminishing return on continued pre-training. This could be used as a minimal bound to consider transitioning to the fine-tuning phase.
>
> **“how these findings can inform the developing of LLMs”**
> We appreciate the reviewer’s concern. However, the primary motivation of our study is not to propose immediate improvements to language model development, but rather to deepen our understanding of the effects of fine-tuning on model behavior.
> However, we have a few potential follow-up ideas in mind:
> 0. The answer to the question "Can you elaborate on potential signals or metrics during pre-training that could indicate an optimal point to stop pre-training and begin fine-tuning?"
>
> 1. Finding a balance point between the cost of training and the final performance.
>
> 2. Under a use case, achieve the best performance possible with an appropriate combination of pre-training and fine-tuning.

---

### Official Review · Reviewer_S7Wz · 2024-11-06

**Soundness:** 3
**Presentation:** 2
**Contribution:** 2
**Rating:** 5
**Confidence:** 3

**Summary:**

This paper investigates the relationship between pre-training and fine-tuning in large language models by fine-tuning multiple intermediate pre-trained model checkpoints. The authors aim to understand how models develop during pre-training and how this affects their performance after fine-tuning on downstream tasks. The main contributions include empirical findings that continual pre-training improves models in ways only revealed after fine-tuning, that fine-tuning benefits tasks not learned during pre-training, that fine-tuning can cause forgetting of previously known tasks, and that prompt sensitivity after fine-tuning can be mitigated with more pre-training.

**Strengths:**

1. The paper addresses an under-explored area by empirically studying the interplay between pre-training and fine-tuning stages in language model development.
2. It provides valuable insights that can inform more efficient training strategies, such as early stopping in pre-training when fine-tuning yields better results.
3. The study is thorough, involving experiments on 18 datasets across various tasks, enhancing the validity of the conclusions.

**Weaknesses:**

1. The study focuses on a single, relatively small model (OLMo-1B), which may limit the applicability of the findings to larger models or different architectures.
2. Due to the scarcity of models with available pre-training checkpoints, the conclusions are based on limited data, potentially affecting the robustness of the results.
3. The paper primarily analyzes downstream performance without deep exploration of model internals or theoretical underpinnings of the observed phenomena.
4. The benchmark datasets (flan-style) seem too simple and out of date for modern LLMs. For example, MT-bench, alpaca-eval, and arena-hard.

**Questions:**

1. Could you provide more details on the selection criteria for the datasets and how they might influence the observed dichotomy between tasks learned during pre-training and those requiring fine-tuning?
2. How do you anticipate your findings would generalize to larger models or different architectures, given that your study was conducted on a relatively small model?
3. Can you elaborate on potential signals or metrics during pre-training that could indicate an optimal point to stop pre-training and begin fine-tuning?

---

> ### Author Response · Authors · 2024-11-19
>
> **Models and Experiments**
>
> Our study includes two models, OLMo-1b and Llama3-8B, and results on both models reach the same conclusions. Experiments that required pre-training checkpoints only include OLMo since Llama did not release checkpoints. We conducted an exhaustive search for pre-training checkpoints, including contacting several model authors. We are aware of only a few other models with checkpoints, which all have issues. 1) TinyLlama fixed a token problem mid-pre-training, which changed model behavior during pre-training. 2) RedPjama, which we experimented with extensively but performed poorly across all of our fine-tuning experiments. 3) Baichuan2, which is multi-lingual (introduces other issues) and is relatively unknown. 4) LLM360, which has a staged pre-training process that deviates from most other models.
>
> While we acknowledge the limitations of our study, we believe it offers valuable insights into a relatively unexplored area of the training process. Our findings highlight an important starting point that can inspire and guide future work. We hope this study demonstrates the value of pre-training checkpoints and encourages model builders to make them more widely available.
>
> To the best of our knowledge, no prior work has explored pre-training checkpoint experiments. Given the lack of alternative models to evaluate and the substantial resources we dedicated—over 1100 A100 GPU hours—we believe this work represents a significant step forward in this domain. We hope it underscores the feasibility of such research for the academic community, even within resource constraints.
>
>
> **“The benchmark datasets (flan-style) seem too simple and out of date for modern LLMs. For example, MT-bench, alpaca-eval, and arena-hard.”**
>
> Even though our datasets seem simple, the model does poorly on them during pre-training. Furthermore, our focus in on supervised fine-tuning, not instruction following. The datasets (e.g., MT-Bench, Alpaca-Eval, Arena-Hard) are specifically designed with instructions, which are orthogonal to our core research questions. Instruction-heavy benchmarks introduce an additional confounding factor—namely, the model's instruction-following ability—rather than the core task-solving abilities we aim to study.
> That said, we agree that exploring the intersection of fine-tuning and instruction-following ability is an interesting direction for future work, and we hope our current findings can serve as a foundation for such analyses.
>
> **Answer to reviewer’s questions:**
>
> _**“Could you provide more details on the selection criteria for the datasets and how they might influence the observed dichotomy between tasks learned during pre-training and those requiring fine-tuning?”**_
> The datasets are selected based on potential data contamination. In addition to the datasets that we are sure were not contaminated, we also select the datasets and tasks (summarization, NLI, QA) that have been prevalent in NLP research.
>
> _**“How do you anticipate your findings would generalize to larger models or different architectures, given that your study was conducted on a relatively small model?”**_
>
> As mentioned in Section xxx, the trend we found with Llama-8b is consistent with the findings with OLMo-1B. We believe the same finding is relatively generable for language models that are trained in a one-stage manner, which includes most Llama families, OLMo version 1, T5 family, etc.
>
> _**“Can you elaborate on potential signals or metrics during pre-training that could indicate an optimal point to stop pre-training and begin fine-tuning?”**_
>
> Emprically, a practical approach is to use a set of validation datasets (for example, datasets in the first exp section) that have been examined to improve throughout pre-training. Those datasets do not require fine-tuning, but they can approximately entail the time when pre-training is sufficient.
> Once the performance on these validation sets plateaus or stops improving, it generally signals a diminishing return on continued pre-training. This could be used as a minimal bound to consider transitioning to the fine-tuning phase.

---

### Note · Authors · 2024-12-16

I have read and agree with the venue's withdrawal policy on behalf of myself and my co-authors.